# Performance of the FebriDx Rapid Point-of-Care Test for Differentiating Bacterial and Viral Respiratory Tract Infections in Patients with a Suspected Respiratory Tract Infection in the Emergency Department

**DOI:** 10.3390/jcm13010163

**Published:** 2023-12-27

**Authors:** Kirby Tong-Minh, Katrijn Daenen, Henrik Endeman, Christian Ramakers, Diederik Gommers, Eric van Gorp, Yuri van der Does

**Affiliations:** 1Department of Emergency Medicine, Erasmus University Medical Center, 3015 GD Rotterdam, The Netherlands; k.tong-minh@erasmusmc.nl (K.T.-M.); y.vanderdoes@umcutrecht.nl (Y.v.d.D.); 2Department of Viroscience, Erasmus University Medical Center, 3015 GD Rotterdam, The Netherlands; e.vangorp@erasmusmc.nl; 3Department of Intensive Care, Erasmus University Medical Center, 3015 GD Rotterdam, The Netherlands; h.endeman@erasmusmc.nl (H.E.); d.gommers@erasmusmc.nl (D.G.); 4Department of Clinical Chemistry, Erasmus University Medical Center, 3015 GD Rotterdam, The Netherlands; c.ramakers@erasmusmc.nl; 5Department of Internal Medicine, Erasmus University Medical Center, 3015 GD Rotterdam, The Netherlands

**Keywords:** infectious diseases, emergency medicine, point-of-care

## Abstract

FebriDx is a rapid point-of-care test combining qualitative measurements of C-reactive protein (CRP) and Myxovirus Resistance Protein A (MxA) using a disposable test device to detect and differentiate acute bacterial from viral respiratory tract infections. The goal of this study was to investigate the diagnostic accuracy of FebriDx in patients with suspected respiratory tract infections in the emergency department (ED). This was an observational cohort study, performed in the ED of an academic hospital. Patients were included if they had a suspected infection. The primary outcome was the presence of a bacterial or viral infection, determined by clinical adjudication by an expert panel. The sensitivity, specificity, and positive and negative predictive value of FebriDx for the presence of bacterial versus non-bacterial infections, and viral versus non-viral infections were calculated. Between March 2019 and November 2020, 244 patients were included. A bacterial infection was present in 41%, viral infection was present in 24%, and 4% of the patients had both viral and bacterial pathogens. FebriDx demonstrated high sensitivity in the detection of bacterial infection (87%), high NPV (91%) to rule out bacterial infection, and high specificity (94%) for viral infection in patients with a suspected infection in the ED.

## 1. Introduction

In the emergency department (ED), viral and bacterial respiratory infections need to be differentiated in order to guide proper management, such as the empirical administration of antibiotics, antibiotic stewardship, and isolation of patients with viral infections [1].

The current diagnostic workup to determine the origin of infection includes sampling of cultures and molecular testing, most often via polymerase chain reaction (PCR). However, it may take hours up to multiple days before the results are available, and PCR cannot differentiate acute infection from colonization. Standard laboratory tests in the ED often include C-reactive protein (CRP), white blood cell count (WBC), and procalcitonin (PCT) [2,3]. These laboratory tests are biomarkers of disease severity and may also aid in distinguishing viral from bacterial infections [4,5]. However, these biomarkers are not always specific enough to guide clinical decision-making on differentiating between viral and bacterial infections [6]. Myxovirus Resistance Protein A (MxA) is a key protein in the interferon (IFN) type-1-regulated antiviral response, and elevated levels are detectable 24–48 h after IFN induction by the innate immune system [7]. MxA has antiviral properties by inhibiting the transcriptional and replicative functions of a wide range of RNA viruses, such as influenza, parainfluenza, and coxsackieviruses [8]. MxA is not induced by pathways activated in bacterial infections, such as those leading to the production and release of IFN-gamma, interleukin-1, and tumor necrosis factor alpha. Therefore, MxA is a potential selective marker of viral infections. In pediatric patients, MxA measurements in whole blood could differentiate patients with viral infections from non-infections and bacterial infections in the ED [9,10]. In ED patients suspected of COVID-19 infection, higher MxA levels were reported in patients with COVID-19 compared to those with other infections [10,11]. The integration of CRP for identifying bacterial infections and MxA for identifying viral infections into a unified diagnostic test holds the potential to reliably distinguish between bacterial and viral infections.

Rapid point-of-care tests (POCTs) for different biomarkers and pathogens are available to speed up the diagnostic workup in the ED [12,13]. FebriDx is a POCT that combines the qualitative measurement of CRP and MxA in a disposable single-use test device with a turnaround time of 10 min [14] intended for immunocompetent patients with a respiratory tract infection (RTI). Previous studies showed that FebriDx is able to detect and differentiate bacterial and viral infections in patients with a suspected RTI in the ED [15]. Its use in a broader population of patients with fewer specific respiratory symptoms and multiple comorbidities, such as immunodeficient patients, in the ED has not yet been investigated.

Therefore, the goal of this study was to investigate the diagnostic accuracy, defined as the sensitivity, specificity, positive predictive value (PPV), and negative predictive value (NPV) of FebriDx in this patient population.

## 2. Materials and Methods

### 2.1. Study Design

This study was performed as a sub study in the FORESEEN study, a prospective observational study that aimed to investigate and validate multiple biomarkers in a heterogeneous cohort of patients with any kind of suspected infection in the ED. Patients were included in the ED of Erasmus University Medical Center, an academic hospital in the Netherlands. Subjects were included between the 15th of March 2019 until the 27th of November 2020. This study was part of a biobank study, which was approved by the medical ethical committee of Erasmus MC under protocol number MEC-2017-417. Additional approval for the enrollment of COVID-19 patients within the same study protocol was given on 12 March 2020.

### 2.2. Inclusion and Exclusion Criteria

Patients were eligible for inclusion in the FORESEEN study if they were 18 years or older, if any cultures were taken or if antibiotics were prescribed in the ED, and they gave written informed consent. Patients were excluded if they did not speak Dutch or did not have a permanent place of residence. For this sub study, patients were included when they were enrolled in the FORESEEN study and either had respiratory symptoms, or when a viral throat swab was obtained as a surrogate criterion for a suspected respiratory infection by the physician.

### 2.3. FebriDx Testing

FebriDx is a rapid, qualitative, single-use, disposable, whole blood immunoassay with a turnaround time of 10 min. FebriDx requires a single droplet of capillary whole blood obtained via a finger prick, which is incorporated into the device. FebriDx provides qualitative results for elevated levels of CRP (≥20 mg/L) and MxA (≥40 ng/mL). According to the manufacturer’s protocol, a positive CRP line (black) with a negative MxA (Figure 1A) line should be interpreted as a bacterial infection, and a positive MxA line (red), with either a positive or negative CRP line should be interpreted as a viral infection (Figure 1B,C). Negative CRP and MxA lines with only a positive control line (blue) should be interpreted as negative (Figure 1D). A test is only considered valid when the control line is colored blue.

The study personnel were trained by the manufacturer of FebriDx to perform the test. The study personnel performed the FebriDx test during the ED visit of the included patients. After collecting the blood sample, the FebriDx test was done according to the manufacturer’s instructions.

The results of the FebriDx test were recorded after 10 min. The treating physicians were blinded to the results.

### 2.4. Data Collection

Patient data were recorded during the ED visit, and included demographic data, comorbidities, duration of symptoms, vital signs, routine laboratory tests, and microbiological cultures. Other than routine laboratory testing, additional diagnostic tests were not performed, with the exception of procalcitonin (PCT) measurements, which were performed in batches at the end of the study, and thus the results were not available to the treating physicians during the patient visit. Cultures were ordered by the physician if there was a clinical suspicion of a specific infection. Viral swabs were ordered when a viral RTI was suspected or when a viral infection needed to be ruled out to apply or lift quarantine measures.

The use of immunosuppressive medication, defined as the use of systemic corticosteroids >7.5 mg prednisone equivalent per day, TNF-a inhibitors, or similar or any antirejection medication after organ transplantation, was recorded. An immunodeficiency as comorbidity was defined as any comorbidity that led to a reduced or compromised immune status, such as primary immunodeficiency and hematological malignancies. Patients were considered non-immunocompetent if they either used immunosuppressive medication or had an immunodeficiency as comorbidity. Laboratory and microbiological tests were only available when ordered by the treating physician. Thirty days following the initial ED visit, data on patient disposition after the ED visit including mortality and infection status were collected via questionnaire and telephone interview.

### 2.5. Outcome

The primary outcome of this study was the presence of a bacterial or viral infection during the ED visit. The type of infection was determined by clinical adjudication using an expert panel of two physicians trained in infectious diseases. A predefined set of criteria was used to determine if patients had a bacterial, viral, or no infection (Appendix A). These criteria are based on the study of Bauer et al. who conducted a study in a similar setting [16]. Both infection types were classified into four categories: no infection, unlikely infection, likely infection, confirmed infection. The expert panel independently assessed all clinical data, including vital signs, radiological, laboratory, PCT values, and microbiological results and compared the result. Discrepancies were discussed until consensus was met. If no consensus was reached, a third physician acted as referee. The expert panel was blinded to the FebriDx results.

Following this, the four categories were transformed into a binary outcome combining the confirmed and likely infections into (bacterial/viral) infection present, and unlikely and no infection into (bacterial/viral) infection absent. Bacterial and viral infection status was scored separately, and as a result patients could be classified as non-infected, only bacterial infection, only viral infection, and bacterial and viral coinfection.

### 2.6. Statistical Analysis

Normally distributed variables were reported as mean with standard deviation (SD) with non-normally distributed variables as median with an interquartile range (IQR). Multiple imputation was used for handling missing data. For the primary analysis, the diagnostic accuracy of FebriDx to detect and differentiate a bacterial from viral infection was calculated in the entire cohort and separately in the group of immunocompetent patients with an acute RTI, which is the intended patient group as defined by the manufacturer. The sensitivity, specificity, positive predictive value, and negative predictive value of FebriDx for the presence of bacterial versus non-bacterial infections, and viral versus non-viral infections were calculated, with 95% confidence intervals (95%CI). For the secondary analysis, the diagnostic accuracy of FebriDx was calculated for the binary outcome of confirmed bacterial or viral infection versus no bacterial or viral infection. In this analysis, the cases with an unconfirmed infection (unlikely and likely infections) were excluded, because the clinical presentation of these patients can be ambiguous, and adjudication can be subjective, resulting in different clinical interpretations between physicians. To investigate the performance of FebriDx in immunocompromised patients, we analyzed the diagnostic accuracy of FebriDx in patients with either an immunodeficiency or patients using immunosuppressive medication. Because this study was a sub study of the FORESEEN study, we used a convenience sample size of all patients who were eligible for FebriDx testing within the FORESEEN study. R version 4.1.0 was used for the statistical analyses.

## 3. Results

The FORESEEN study was conducted between the 15 March 2019 and 27 November 2020. FebriDx tests were performed between the 27 December 2019 and 27 November 2020. In this period, a total of 1389 patients were screened for eligibility for inclusion of which 489 patients were included in the FORESEEN study. A valid FebriDx test was performed in 244 patients, of which 114 were immunocompetent patients with symptoms of an acute RTI as per the FebriDx test’s intended use (Figure 2).

There were missing data in 17/244 (7%) patients. In Appendix A, an overview of missing data per variable is shown.

A total of 228/244 (93%) patients had one or more comorbidities, a total of 70/244 (29%) patients used immunosuppressive medication, and 10/244 (4%) patients had an immunodeficiency. In 73/244 (30%) patients, FebriDx was performed because a viral swab was taken, in the absence of respiratory symptoms, for example for COVID-19 screening. The baseline characteristics for all included patients and the immunocompetent RTI cohort are presented in Table 1. The median age of the study population was 59 years (IQR: 21 years). After the ED visit, 98/244 (40%) patients were discharged home, 145/244 (59%) patients were admitted to the general ward, and 1/244 (<1%) patients was admitted to the ICU. Within 30 days after their ED visit, 7/244 (3%) patients died. Symptom onset ranged from 1–43 days with a median time of onset of symptoms of 3 days (IQR: 4).

### 3.1. Clinical Adjudication

After clinical adjudication of the immunocompetent patients with an ARI, 13/114 (11%) patients were classified as confirmed bacterial, 25/114 (22%) patients as likely bacterial, 27/114 (24%) patients as unlikely bacterial, and 49/114 (43%) patients as no bacterial infection. Clinical adjudication classified 33/114 (29%) patients as confirmed viral, 6/114 (5%) patients as likely viral, 9/114 (8%) patients as unlikely viral, and 66/114 (58%) patients as no viral infection. Following conversion to the binary outcome of bacterial or viral infection present or absent, in 38/114 (33%) patients, a bacterial infection was present versus 76/114 (67%) patients without a bacterial infection and 39/114 (34%) patients where a viral infection was present versus 75/114 (66%) patients without a viral infection. In 6/114 (5%) patients, there were both a bacterium and virus co-detected. The outcome of the clinical adjudication in the total cohort is shown in Figure 2. The list of pathogens detected is provided in Table 2.

### 3.2. Diagnostic Accuracy FebriDx in Immunocompetent Patients with Symptoms of Acute RTI

In immunocompetent patients with an acute ARI, the CRP line alone was positive in 58/114 (51%) patients, the MxA line alone was positive in 4/114 (4%) patients, and both CRP and MxA lines were positive in 24/114 (21%) patients. In 28/114 (25%) patients, both CRP and MxA lines were negative. Following the manufacturer’s interpretation scheme, this resulted in FebriDx classifying 58/114 (51%) patients as having a bacterial infection, 28/114 (25%) patients as having a viral infection, and 24/114 (21%) patients as negative. FebriDx had a sensitivity of 87% (95%CI 72–96%) and specificity of 67% (95%CI 55–77%) to detect bacterial infections with a PPV of 57% (95%CI 43–70%) and NPV of 91% (95%CI 80–97%). FebriDx had a sensitivity of 56% (95%CI 40–72%) and specificity of 92% (95%CI 83–97%) to detect viral infection with a PPV of 79% (95%CI 59–92%) and NPV of 80% (95%CI 70–92%).

### 3.3. Diagnostic Accuracy of FebriDx in the Total Cohort

In the total cohort, FebriDx classified 134/244 (55%) patients as having a bacterial infection, 39/244 (16%) patients as having a viral infection, and 71/244 (29%) patients as negative.

FebriDx had a sensitivity of 87% (95%CI 78–93%) and specificity of 64% (95%CI 56–72%) to detect bacterial infections with a PPV of 60% (95%CI 51–68%) and NPV of 89% (95%CI 82–94%). FebriDx had a sensitivity of 49% (95%CI 36–63%) and specificity of 94% (95%CI 90–97%) to detect viral infection with a PPV of 72% (95%CI 56–85%) and NPV of 85% (95%CI 80–90%).

### 3.4. Diagnostic Accuracy of FebriDx in Other Subgroups

When excluding patients with an unconfirmed infection (i.e., no pathogen identified with the host response and clinical data not conclusive), 62/114 (57%) patients remained whom were classified as confirmed bacterial and non-bacterial and 98/114 (86%) patients as confirmed viral and non-viral. FebriDx had a sensitivity of 85% (95%CI 55–98%) and specificity of 71% (95%CI 57–83%) with a PPV of 44% (95%CI 24–65%) and a NPV of 95% (95%CI 82–99%) to detect bacterial infections when analyzing only the confirmed bacterial infections. Viral sensitivity was 58% (95%CI 39–75%) and specificity 97% (95%CI 89–100%) with a PPV of 90% (95%CI 70–99%) and NPV of 82% (95%CI 71–90%) when the analysis was limited to confirmed viral infections.

The analysis of the diagnostic accuracy of FebriDx in the subgroup of immunocompromised is shown in Table 3.

## 4. Discussion

In this study we investigated the diagnostic accuracy of the POCT FebriDx to detect and differentiate viral from bacterial RTIs in the ED in patients with a suspected RTI. The combination of CRP and MxA in a single test has the potential of reliably differentiating bacterial from viral infections. In the current study, FebriDx showed a sensitivity of 87% (95%CI 72–96%) and specificity of 67% (95%CI 55–77%) to detect bacterial infections and a sensitivity of 56% (95%CI 40–72%) and specificity of 92% (95%CI 83–97%) to detect viral infections in immunocompetent patients with symptoms of acute RTI.

With a specificity of 92% in identifying viral infections, FebriDx could be used to aid in decisions to treat with, or withhold, antibiotics. In patients that are identified as having viral infections using FebriDx, a consideration could be to withhold early empiric antibiotic treatment when no bacterial coinfection is suspected. Due to the sensitivity of 56% for identifying viral infections that we found in this study, FebriDx may have limited clinical significance in relation to decisions concerning the allocation of isolation rooms for patients with a suspected viral RTI. This is in contrast to other studies, performed during the COVID-19 pandemic, that showed that FebriDx can be used as triage tool to aid isolation decisions [17,18].

Our study showed that FebriDx was able to identify bacterial infections, both in patients with a confirmed infection and patients in which the infection status was uncertain (likely and unlikely infections). Application of FebriDx during triage could hypothetically lead to earlier identification of patients with bacterial infections. Further studies should investigate this hypothesis, and whether the use of FebriDx leads to earlier treatment with antibiotics and reduced mortality in patients with a bacterial infection. Based on other studies, we expect that implementing FebriDx will lead to a more efficient patient flow in the ED when bacterial infections are identified during triage [17,18,19].

Our results differ from previous studies that investigated FebriDx in regard to several points [14,15,20,21,22]. In general, the diagnostic accuracy in identifying viral infections was lower in our study than described in the literature [15,20,21]. This difference may be attributable to differences in clinical adjudication methods as well as a less comorbid, severely ill patient population in the aforementioned publications. Furthermore, PCR can detect viruses known to colonize the respiratory tract whilst not causing infection, blood cultures might detect pathogens not directly associated with the respiratory tract, and only common pathogens associated with respiratory infection were determined to be causative agents in previous studies, which may offer an explanation as to why our results differ from those of previous studies. It should be noted that five viral false negatives were due to virus detection via PCR without an associated host immune response (elevated PCT or WBC) or a measure of viral load (cycle threshold), and thus it is not possible to determine whether these were active infections or related to viral shedding from a past infection.

We tested FebriDx in a heterogeneous cohort of patients without limiting the population to that intended for use of the device. Patients were included when having a suspected infection, without selecting participants based on the type and duration of symptoms, a history of fever, and measured hyperthermia in the ED, as was the case in other studies [15,22]. In our study, this resulted in a mixed group without specific symptoms, where the final diagnosis was not always a RTI, and in several instances, not an infection but a complication of underlying conditions such as heart failure, cystic fibrosis, or cancer. Furthermore, these patients often presented with symptoms from multiple organ systems, requiring additional sampling besides a throat swab, resulting in positive urine and stool cultures. Furthermore, in our real-world clinical study, only diagnostic tests that were ordered by the treating physicians were available during clinical adjudication. This was in contrast to the adjudication methods used in previous studies that investigated FebriDx [15,18,19,21,22]. As a result, cultures were missing in 6/244 (2%) and molecular testing of respiratory viral infections was missing in 80/244 (33%) of the patients; therefore, clinical adjudication was based on the clinical course alone. This may have led to misclassification during clinical adjudication. However, it does reflect the real-world performance of FebriDx as it would be used in clinical practice, where a full diagnostic workup is also not always performed.

In contrast to other studies that investigated FebriDx, we included patients with an immunodeficiency and patients that used immunosuppressive medication [15,20,22]. MxA is produced in response to viral infections and upregulated by type-1 interferons. In immunocompromised patients, this host response is reduced, leading to a reduced production of MxA [23]. In our study population, 32% of the patients were immunocompromised either due to use of an immunosuppressive medication or an immunodeficiency. In a sub analysis of these patients, we did see that the sensitivity of the identification of viral infections was lower than in immunocompetent patients. Although this analysis was conducted in a small subgroup of 50 patients, these data suggest that the performance of FebriDx is reduced in the immunocompromised patients enrolled in our study.

### Limitations

This study was performed in an academic medical center in the Netherlands, where patients often have multiple comorbidities and use immunosuppressive medication after organ transplantation, which reduces the generalizability of the results. Furthermore, patients visiting an academic hospital in the Netherlands are always referred by their general practitioner. This results in a higher hospital admission rate compared to a healthcare system where patients primarily visit the ED when they have respiratory symptoms. If validated in a multicenter setting also including non-academic urban hospitals, the results will be more generalizable to a general ED population.

The study started before the outbreak of the COVID-19 pandemic, and enrollment continued throughout the first two waves of COVID-19 in 2020. This resulted in a different patient population than other studies published before 2020, and the results may not be comparable.

Similar to other studies that investigated the diagnostic accuracy of biomarkers or diagnostic tests for bacterial or viral infections, clinical adjudication is limited by the chosen adjudication criteria and the available clinical data and is never perfect. In the absence of a gold standard for determining the infection status, misclassification of the final infection type is inevitable and unavoidable.

In our study, children were not included, constituting a limitation given the pressing need for improved diagnostic tools in this specific patient group. Additionally, beyond the future evaluation of FebriDx in this patient group, the exploration of upcoming tools incorporating multiple pathways derived from transcriptomic analyses may be beneficial [24].

The current study was observational; therefore, future interventional studies should investigate whether routinely using FebriDx in the ED results in reduced antibiotic prescriptions when employed to guide antibiotic treatment. In addition, future investigations should explore the potential contribution of FebriDx to enhance patient flow in the ED when employed for guiding isolation measures.

## 5. Conclusions

The POCT FebriDx demonstrated a high sensitivity in the detection of bacterial infection, high NPV to rule out bacterial infection, and high specificity for viral infection in patients with a suspected infection in an academic hospital.

## Figures and Tables

**Figure 1 jcm-13-00163-f001:**
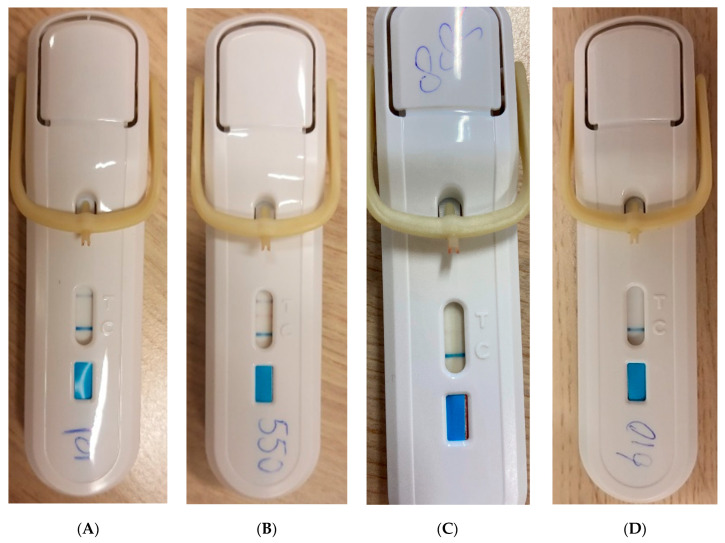
Different results of the FebriDx test. (**A**) FebriDx with positive CRP line (black) and positive control line (blue)—Bacterial Infection. (**B**) FebriDx with positive CRP line (black), MxA line (red), and control line (blue)—Viral Infection. (**C**) FebriDx with positive MxA line (red), and control line (blue)—Viral Infection. (**D**) FebriDx with negative CRP and MxA line and positive control line (blue)—Negative.

**Figure 2 jcm-13-00163-f002:**
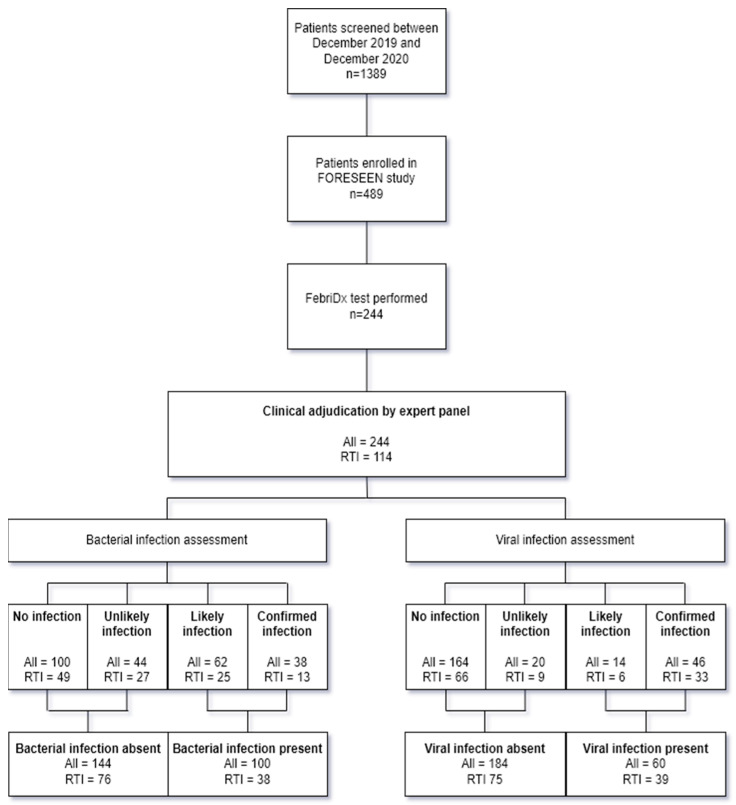
Flowchart of included patients and outcome of clinical adjudication in total cohort and subgroup of immunocompetent patients with an acute RTI. RTI: Respiratory tract infection.

**Table 1 jcm-13-00163-t001:** Baseline characteristics.

		Non-Infected		Only Bacterial Infection		Only Viral Infection		Detection of Bacterial and Viral Pathogens		All	
		Total Cohort	Immunocompetent Acute RTI	Total Cohort	Immunocompetent Acute RTI	Total Cohort	Immunocompetent Acute RTI	Total Cohort	Immunocompetent Acute RTI	Total Cohort	Immunocompetent Acute RTI
		(N = 95)	(N = 43)	(N = 90)	(N = 32)	(N = 49)	(N = 33)	(N = 10)	(N = 6)	(N = 244)	(N = 114)
**Demograpic data**											
Age	median (IQR)	57.5 (28.0)	63.0 (27.5)	64.5 (15.0)	64.5 (13.8)	57.0 (16.5)	57.0 (9.00)	51.5 (6.25)	50.5 (6.25)	59.0 (21.0)	59.0 (19.8)
Sex: male	n (%)	45 (44.1%)	23 (53.5%)	45 (54.9%)	16 (50.0%)	32 (64.0%)	19 (57.6%)	6 (60.0%)	5 (83.3%)	128 (52.5%)	63 (55.3%)
Comorbidity: Cardiovascular disease	n (%)	39 (38.2%)	16 (37.2%)	42 (51.2%)	15 (46.9%)	22 (44.0%)	13 (39.4%)	4 (40.0%)	2 (33.3%)	107 (43.9%)	46 (40.4%)
Comorbidity: Central nervous system diseases	n (%)	5 (4.9%)	3 (7.0%)	8 (9.8%)	3 (9.4%)	1 (2.0%)	1 (3.0%)	0 (0%)	0 (0%)	14 (5.7%)	7 (6.1%)
Comorbidity: pulmonary disease	n (%)	40 (39.2%)	14 (32.6%)	26 (31.7%)	14 (43.8%)	18 (36.0%)	12 (36.4%)	4 (40.0%)	2 (33.3%)	88 (36.1%)	42 (36.8%)
Comorbidity: Diabetes Mellitus	n (%)	15 (14.7%)	6 (14.0%)	23 (28.0%)	5 (15.6%)	11 (22.0%)	8 (24.2%)	4 (40.0%)	1 (16.7%)	53 (21.7%)	20 (17.5%)
Comorbidity: Renal disease	n (%)	7 (6.9%)	2 (4.7%)	17 (20.7%)	3 (9.4%)	7 (14.0%)	2 (6.1%)	2 (20.0%)	0 (0%)	33 (13.5%)	7 (6.1%)
Comorbidity: Liver disease	n (%)	4 (3.9%)	1 (2.3%)	5 (6.1%)	2 (6.3%)	4 (8.0%)	1 (3.0%)	1 (10.0%)	0 (0%)	14 (5.7%)	4 (3.5%)
Comorbidity: Malignancy	n (%)	30 (29.4%)	16 (37.2%)	31 (37.8%)	16 (50.0%)	7 (14.0%)	4 (12.1%)	1 (10.0%)	1 (16.7%)	69 (28.3%)	37 (32.5%)
Comorbidity: Immunodeficiency	n (%)	7 (6.9%)	43 (100%)	1 (1.2%)	32 (100%)	1 (2.0%)	33 (100%)	1 (10.0%)	6 (100%)	10 (4.1%)	114 (100%)
Comorbidity: Auto-immune diseases		18 (17.6%)	2 (4.7%)	15 (18.3%)	3 (9.4%)	5 (10.0%)	3 (9.1%)	2 (20.0%)	0 (0%)	40 (16.4%)	8 (7.0%)
Comorbidity: Solid organ transplantation		13 (12.7%)	3 (7.0%)	16 (19.5%)	0 (0%)	12 (24.0%)	2 (6.1%)	1 (10.0%)	0 (0%)	42 (17.2%)	5 (4.4%)
**Vital signs**											
Heartrate	Mean (SD)	90.2 (16.6)	93.1 (19.2)	98.5 (19.7)	99.7 (17.5)	96.3 (16.1)	94.4 (15.8)	101 (22.5)	103 (24.0)	94.7 (18.2)	95.8 (18.1)
Systolic blood pressure	Mean (SD)	141 (22.9)	139 (21.3)	135 (24.1)	134 (21.7)	137 (20.0)	136 (19.3)	132 (17.0)	142 (20.7)	138 (22.6)	137 (20.7)
Diastolic blood pressure	Mean (SD)	85.6 (14.9)	82.9 (16.7)	82.0 (15.9)	81.5 (14.0)	84.0 (13.9)	83.2 (14.1)	84.4 (18.7)	84.2 (8.45)	84.0 (15.2)	82.6 (14.8)
Temperature (Celcius)	Mean (SD)	37.0 (0.706)	37.1 (0.723)	37.6 (0.820)	37.7 (0.875)	37.5 (0.807)	37.4 (0.851)	37.7 (0.650)	37.7 (0.417)	37.4 (0.810)	37.4 (0.822)
Respiratory rate	Mean (SD)	19.7 (5.09)	19.4 (5.09)	21.2 (6.46)	23.0 (7.49)	21.1 (5.46)	22.1 (5.44)	22.8 (5.01)	22.0 (4.38)	20.6 (5.69)	21.4 (6.05)
SpO_2_	median (IQR)	95.2 (8.93)	96.0 (2.00)	95.3 (3.22)	95.5 (2.25)	95.0 (5.22)	95.0 (3.00)	95.0 (3.62)	96.5 (1.00)	95.2 (6.53)	96.0 (3.00)
**Laboratory tests**											
CRP	median (IQR)	36.3 (53.9)	10.0 (41.7)	131 (112)	104 (132)	56.0 (67.5)	33.0 (51.2)	82.9 (65.6)	70.0 (95.5)	74.1 (90.9)	34.5 (93.9)
WBC	median (IQR)	12.0 (40.9)	8.40 (4.45)	103 (135)	12.2 (6.08)	42.5 (52.5)	7.10 (3.10)	81.0 (112)	6.30 (5.68)	42.5 (92.7)	8.60 (5.58)
PCT	median (IQR)	0.0500 (0.0775)	0.0600 (0.160)	0.290 (0.730)	0.150 (0.475)	0.0700 (0.0675)	0.0600 (0.0600)	0.365 (0.518)	0.140 (0.163)	0.100 (0.250)	0.0900 (0.215)
**Disposition and severity**											
Discharge home	n (%)	61 (64.2%)	25 (58.1%)	15 (16.4%)	5 (15.6%)	18 (36.7%)	11 (33.3%)	4 (36.4%)	3 (50.0%)	98 (40.2%)	44 (38.6%)
Admission to general ward	n (%)	34 (35.8%)	18 (41.9%)	74 (83.1%)	27 (84.4%)	30 (61.2%)	21 (63.6%)	7 (63.6%)	3 (50.0%)	145 (59.4%)	69 (60.5%)
Admission to intensive care unit	n (%)	0 (0%)	0 (0%)	0 (0%)	0 (0%)	1 (2.0%)	1 (3.0%)	0 (0%)	0 (0%)	1 (0.4%)	1 (0.9%)
Admission duration	median (IQR)	0 (3.00)	0 (3.50)	6.00 (6.00)	8.00 (7.25)	5.00 (8.00)	5.00 (8.00)	3.00 (8.00)	1.50 (6.75)	3.00 (7.00)	3.00 (8.00)
30-day mortality	n (%)	1 (1.1%)	1 (2.3%)	5 (5.6%)	4 (12.5%)	1 (2.0%)	0 (0%)	0 (0%)	0 (0%)	7 (2.9%)	5 (4.5%)

**Table 2 jcm-13-00163-t002:** Microbiological results.

All Patients N = 244		RTI Immunocompetent Patients N = 114	
Microbiological Results	Number of Patients	Microbiological Results	Number of Patients
Blood Cultures		Blood Cultures	
*Escherichia coli (E. coli)* *Staphylococcus epidermidis (S. epidermidis)* *Klebsiella pneumoniae (K. pneumoniae)* *Staphylococcus hominis (S. hominis)* *Staphylococcus aureus (S. aureus)* *Staphylococcus capitis (S. capitis)* *Streptococcus pneumoniae (S.pneumoniae)* *Streptococcus Hemolyticus* *Micrococcus luteus (M. luteus)* *Parabacteroides distasonis* *Rothia mucilaginosa* *Stenotrophomonas maltophilia*	4 3 3 2 1 1 1 2 1 1 1 1	*Escherichia coli (E. coli)* *Staphylococcus epidermidis (S. epidermidis)* *Klebsiella pneumoniae (K. pneumoniae)* *Staphylococcus hominis (S. hominis)* *Staphylococcus aureus (S. aureus)* *Staphylococcus capitis (S. capitis)* *Streptococcus pneumoniae (S. pneumoniae)* *Streptococcus Hemolyticus* *Micrococcus luteus (M. luteus)* *Parabacteroides distasonis* *Rothia mucilaginosa* *Stenotrophomonas maltophilia*	0 2 0 2 0 1 1 1 0 0 1 0
**Urine antigen testing**		**Urine antigen testing**	
*Streptococcus pneumoniae (S. pneumoniae)*	1	*Streptococcus pneumoniae (S. pneumoniae)*	0
**Sputum cultures**		**Sputum cultures**	
*Staphylococcus aureus (S. aureus)* *Pseudomonas aeruginosa* *Aspergillus fumigatus* *Achromobacter xylosoxidans* *Burkholderia cenocepacia* *Mycoplasma pneumoniae* *Proteus mirabilis* *Serratia marcescens* *Alcaligenes faecalis* *Pneumocystis jirovecii*	3 2 1 1 1 1 1 1 1 1	*Staphylococcus aureus (S. aureus)* *Pseudomonas aeruginosa* *Aspergillus fumigatus* *Achromobacter xylosoxidans* *Burkholderia cenocepacia* *Mycoplasma pneumoniae* *Proteus mirabilis* *Serratia marcescens* *Alcaligenes faecalis* *Pneumocystis jirovecii*	1 1 1 1 1 0 1 1 1 1
**Urine cultures**		**Urine cultures**	
*Escherichia coli (E. coli)* *Actinotignum schaalii* *Enterococcus faecalis (E. faecalis)* *Pseudomonas species* *Acinetobacter baumannii calcoaceticus complex* *Enterococcus cloacae (E. cloacae)* *Staphylococcus aureus (S. aureus)* *Serratia marcescens* *Achromobacter xylosoxidans*	7 2 2 1 1 1 2 1 1	*Escherichia coli (E. coli)* *Actinotignum schaalii* *Enterococcus faecalis (E. faecalis)* *Pseudomonas species* *Acinetobacter baumannii calcoaceticus complex* *Enterococcus cloacae (E. cloacae)* *Staphylococcus aureus (S. aureus)* *Serratia marcescens* *Achromobacter xylosoxidans*	1 1 0 0 0 0 0 0 0
**Other cultures**		**Other cultures**	
*Pseudomona aeruginosa* *Streptococcus pneumoniae (S.pneumoniae) Streptococcus dysgalactiae (S. dysgalactiae)* *Staphylococcus aureus (S. aureus)* *Clostridium difficile (C. difficile)* *Campylobacter jejuni*	3 1 1 1 2 1	*Pseudomona aeruginosa* *Streptococcus pneumoniae (S.pneumoniae) Streptococcus dysgalactiae (S. dysgalactiae)* *Staphylococcus aureus (S. aureus)* *Clostridium difficile (C. difficile)* *Campylobacter jejuni*	3 1 1 0 1 0
**Molecular diagnostics**		**Molecular diagnostics**	
SARS-CoV-2 Rhinovirus Human *metapneumovirus* (hMPV) hMPV + Rhinovirus Influenza (A or B) Adenovirus Bocavirus *Cytomegalovirus* (CMV) *Respiratory syncytial virus* (RSV)	28 7 3 1 5 1 1 2 1	SARS-CoV-2 Rhinovirus Human *metapneumovirus* (hMPV) hMPV + Rhinovirus Influenza (A or B) Adenovirus Bocavirus *Cytomegalovirus* (CMV) *Respiratory syncytial virus* (RSV)	22 3 3 1 4 0 1 0 1

**Table 3 jcm-13-00163-t003:** Secondary analyses of diagnostic accuracy of FebriDx in different subgroups.

Diagnostic Accuracy of FebriDx in RTI Different Subgroups	Infection Type	Number of Patients in Analysis	Sensitivity	Specificity	Positive Predictive Value	Negative Predictive Value
**All patients**	Bacterial infections	244	87% (78–90%)	62% (56–72%)	60% (51–68%)	89% (82–94%)
	Viral infections	244	49% (36–63%)	94% (90–97%)	72% (56–85%)	85% (80–90%)
**RTI immunocompetent**	Bacterial infections	114	87% (72–96%)	67% (55–77%)	57% (43–70%)	91% (80–97%)
	Viral infections	114	56% (40–72%)	92% (83–97%)	79% (59–92%)	80% (70–92%)
**RTI immunocompromised**	Bacterial infections	50	71% (49–87%)	81% (61–93%)	77% (55–92%)	75% (55–92%)
	Viral infections	50	44% (20–70%)	100% (90–100%)	100% (59–100%)	79% (64–90%)
**RTI immunocompetent confirmed infections**	Bacterial infections	57	85% (55–98%)	71% (57–83%)	44% (24–65%)	95% (82–99%)
	Viral infections	98	58% (39–75%)	97% (89–100%)	90% (70–99%)	82% (71–90%)

RTI: Respiratory tract infection.

## Data Availability

The datasets used and/or analyzed during the current study are available from the corresponding author on reasonable request.

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
