# Peer review of "Performance of the FebriDx Rapid Point-of-Care Test for Differentiating Bacterial and Viral Respiratory Tract Infections in Patients with a Suspected Respiratory Tract Infection in the Emergency Department"

_jcm, 2023, doi:10.3390/jcm13010163_

Round 1

Reviewer 1 Report

Comments and Suggestions for Authors

In this manuscript, the authors investigated the diagnostic accuracy of FebriDx in patients with a suspected respiratory tract infection in the emergency department (ED). As a rapid point-of-care-test (POCT) that combines the qualitative measurements  of C-reactive protein (CRP) and Myxovirus Resistance Protein A (MxA) in a disposable single use, FebriDx was  used to detect and differentiate acute bacterial or viral respiratory tract infections. I think this is an important research object for clinlic.  And this work was mainly  an observational cohort study. I have following suggestions for these authors. 

1. The abstract is too long with a lot of data/details, such as sensitivity, specificity, PPV, and NPV. These details/raw data is not necessary in the abstract and should be deleted and summaried/compared in a much more consize way. 

2. Myxovirus Resistance Protein A (MxA) is a key protein in the interferon (IFN) type-1-regulated antiviral response.  It is one of the two most important biomarkers for FebriDx. And it is new compared with previous method.  It should be discussed in more details, and it is much better  some molecular mechansim can be provided. 

3. I suggest the reason for the better perform of FebriDx to distinguish acute bacterial from viral respiratory tract infections should be discussed. 

Author Response

Reviewer 1

  1. The abstract is too long with a lot of data/details, such as sensitivity, specificity, PPV, and NPV. These details/raw data is not necessary in the abstract and should be deleted and summaried/compared in a much more consize way.
  • We adjusted the abstract as suggested by the reviewer.
  1. Myxovirus Resistance Protein A (MxA) is a key protein in the interferon (IFN) type-1-regulated antiviral response. It is one of the two most important biomarkers for FebriDx. And it is new compared with previous method. It should be discussed in more details, and it is much better some molecular mechansim can be provided.
  • We have added more in-depth information about the induction of MxA and its properties in the introduction section (lines 47-58).
  1. I suggest the reason for the better perform of FebriDx to distinguish acute bacterial from viral respiratory tract infections should be discussed.
  • We thank the reviewer for this comment. However, we do not fully understand what the reviewer suggests to adjust, because in our study we showed only a limited added value of FebriDx due to its low sensitivity for viral infections and low specificity for bacterial infections. We are willing to answer and adjust the manuscript if the reviewer can elaborate on this comment.

Reviewer 2 Report

Comments and Suggestions for Authors

Dear authors,

I congratulate you on your findings. Without a doubt, this type of tool helps us to rationalise the management of the various therapeutic resources available to us.

Just a comment on line 60, I suggest providing more background on the usefulness of MxA for the differentiation of viral infections and indicating concrete examples of studies.

I would like to highlight the analysis of the limitations of the study, I think it is excellent but I would suggest at the end to indicate what are the projections for this device.

Kind regards

Author Response

Reviewer 2

Just a comment on line 60 I suggest providing more background on the usefulness of MxA for the differentiation of viral infections and indicating concrete examples of studies.

  • We added information on previously published studies of the clinical use of MxA in the ED in the introduction section (lines 47-58).

I would like to highlight the analysis of the limitations of the study, I think it is excellent but I would suggest at the end to indicate what are the projections for this device.

We added suggestions on future studies for the application of FebriDx in the ED at the end of the discussion section (lines 138-141).

Round 2

Reviewer 1 Report

Comments and Suggestions for Authors

I would like the authors to explain why FebriDx can differentiate bacterial from viral diease with good reliablity. Considering this is only an observational study, I think the revision is fine for publish. 

Author Response

Reviewer 1

I would like the authors to explain why FebriDx can differentiate bacterial from viral diease with good reliablity. Considering this is only an observational study, I think the revision is fine for publish.

  • We have added an extra sentence to the introduction and discussion section why FebriDx has the potential to differentiate bacterial from viral infections (lines 57-59).